# Optimization of Isolation and Transformation of Protoplasts from *Uncaria rhynchophylla* and Its Application to Transient Gene Expression Analysis

**DOI:** 10.3390/ijms24043633

**Published:** 2023-02-11

**Authors:** Yingying Shao, Detian Mu, Limei Pan, Iain W. Wilson, Yajie Zheng, Lina Zhu, Zhiguo Lu, Lingyun Wan, Jine Fu, Shugen Wei, Lisha Song, Deyou Qiu, Qi Tang

**Affiliations:** 1College of Horticulture, Hunan Agricultural University, Changsha 410128, China; 2Guangxi Botanical Garden of Medicinal Plants, Nanning 530023, China; 3CSIRO Agriculture and Food, Canberra, ACT 2601, Australia; 4State Key Laboratory of Tree Genetics and Breeding, Research Institute of Forestry, Chinese Academy of Forestry, Beijing 100091, China

**Keywords:** *Uncaria rhynchophylla*, protoplast isolation, PEG-mediated transfection, subcellular localization, dual-luciferase (Dual-LUC) assays

## Abstract

Protoplast-based engineering has become an important tool for basic plant molecular biology research and developing genome-edited crops. *Uncaria rhynchophylla* is a traditional Chinese medicinal plant with a variety of pharmaceutically important indole alkaloids. In this study, an optimized protocol for *U. rhynchophylla* protoplast isolation, purification, and transient gene expression was developed. The best protoplast separation protocol was found to be 0.8 M D-mannitol, 1.25% Cellulase R-10, and 0.6% Macerozyme R-10 enzymolysis for 5 h at 26 °C in the dark with constant oscillation at 40 rpm/min. The protoplast yield was as high as 1.5 × 10^7^ protoplasts/g fresh weight, and the survival rate of protoplasts was greater than 90%. Furthermore, polyethylene glycol (PEG)-mediated transient transformation of *U. rhynchophylla* protoplasts was investigated by optimizing different crucial factors affecting transfection efficiency, including plasmid DNA amount, PEG concentration, and transfection duration. The *U. rhynchophylla* protoplast transfection rate was highest (71%) when protoplasts were transfected overnight at 24 °C with the 40 µg of plasmid DNA for 40 min in a solution containing 40% PEG. This highly efficient protoplast-based transient expression system was used for subcellular localization of transcription factor UrWRKY37. Finally, a dual-luciferase assay was used to detect a transcription factor promoter interaction by co-expressing UrWRKY37 with a *UrTDC*-promoter reporter plasmid. Taken together, our optimized protocols provide a foundation for future molecular studies of gene function and expression in *U. rhynchophylla*.

## 1. Introduction

Protoplasts are plant cells that have had their cell walls enzymatically removed [1] and are not only a valuable biomaterial but a promising technology platform that has extensive applications in many research fields [2], including protein function, signal transduction [3], transcriptional regulation [4], cellular processes [5,6], and multi-omics analyses [7]. In particular, CRISPR/cas9-mediated genome editing technology can be used to evaluate the efficiency of targeted mutagenesis [8], and plants with required traits can also be regenerated in some species, which can accelerate plant genetic improvement and seed cultivation. The potential role of protoplasts in understanding the mechanism of the formation of functional components in medicinal plants has attracted much attention.

The study of protoplasts emerged in the 1970s and 1980s [9], and there are established methods for protoplast separation in many species. However, efficient protoplast-based systems are still challenging for numerous plant species, due to the different cell wall components that require specific optimization [10]. Enzymatic hydrolysis is often used in protoplast preparation, and the enzyme components and conditions used is the most critical factor affecting protoplast separation and activity [11]. Among many factors, the composition and pH value of the enzyme solution; the temperature, speed, and duration of enzymatic hydrolysis; different enrichment treatments; and the properties of the prefabricated solution all affect the yield and activity of the protoplasts obtained [12].

Protoplast transient transformation and expression systems have been widely used in modern molecular biology analyses such as subcellular localization, protein interactions, and detection of transcription factor activity [13]. Currently, the most common methods for the genetic transformation of plant protoplasts include PEG-mediated and electroporation, with PEG-mediated transfection most widely used due to easy operation, wide applicability, and relatively stable efficiency [14], so it is commonly used for the importation of DNA, RNA, or protein molecules into protoplasts [15]. There are many factors that can affect the PEG-mediated transformation efficiency of protoplasts, among which the key factors are PEG concentration, DNA concentration, the number of protoplasts, and transformation duration [16]. Generally, an efficient transient transfection system requires a stable protoplast separation system that can produce a large pool of highly active protoplasts, which can significantly improve transformation efficiency [17]. However, obtaining high-quality viable protoplasts remains a challenge for *U. rhynchophylla*.

*U. rhynchophylla* is a vine plant in the Rubiaceae family, which has been used in traditional Chinese medicine for thousands of years [18], and the hook stems contain a variety of bioactive components [19]. *U. rhynchophylla* is mainly used for the treatment of convulsions, hypertension, epilepsy, eclampsia, and encephalopathy [20]. With the advances in the pharmacology knowledge of *U. rhynchophylla*, terpenoid indole alkaloids have been identified as important ingredients for their pharmacological effects [21]. However, due to its relatively low alkaloid content, long growth cycle, and high cost in traditional cultivation and breeding, alkaloids derived from *U. rhynchophylla* plants cannot meet the current medical demand. Many studies have shown that specific transcription factors, especially WRKY family members, can simultaneously induce the co-expression of one or more genes or even entire biosynthetic pathways [22]. WRKY transcription factors are one of the largest families of transcription regulators found in plants [23] and are not only intimately involved in various plant growth and development processes but also play a crucial role in plant adaptation to various stress environments [24]. Examples include the overexpression of CrWRKYI in periwinkle, which can dramatically promote the synthesis of terpenoid indole alkaloids, providing genetic resources for the production of *Catharanthus roseus* alkaloids [25]. The expression of ADS and CYP was significantly increased in AaWRKY1-overexpressing transgenic *Artemisia annua* plant [26], and overexpression of OpWRKY6 significantly reduced the accumulation of camptothecin compared to controls in transgenic *Ophiorrhiza pumila* hairy roots [27]. As previously reported, the overexpression of CrWRKY1 in *C. roseus* hairy roots up-regulated several key genes involved in the TIA pathway and increase the accumulation of serpentine compared to the control.

In this study, an effective protoplast separation and transformation protocol were established from the leaves of *U. rhynchophylla*. An optimized method for DNA transformation of *U. rhynchophylla* protoplasts was also developed that enabled transient gene expression subcellular localization analysis of UrWRKY37 to be successfully performed. Additionally, we showed the feasibility of this system to carry out dual-luciferase (Dual-LUC) assays for detecting DNA-protein interactions. The protoplast isolation and transformation method reported here provides a good foundation for the molecular study of *U. rhynchophylla* gene function and understanding of important terpenoid indole alkaloids biosynthesis pathways.

## 2. Results

### 2.1. Protoplast Isolation from U. rhynchophylla Leaves

To establish an efficient protocol for *U. rhynchophylla* mesophyll protoplast isolation and purification, the effects of several conditions were evaluated, and the yield and viability of leaf protoplasts were analyzed. In this study, leaves from 8-week-old *U. rhynchophylla* plantlets were cut into small pieces (0.5–1.0 mm) with scissors for digestion in an enzyme solution.

Isolated protoplasts were stained with FDA to assess their viability and observed under fluorescence microscopy. Healthy and viable protoplasts displayed green fluorescence, while unhealthy or nonviable protoplasts exhibited only chlorophyll autofluorescence (red). It can be seen from Figure 1 that with the yield of protoplasts obtained, viability was calculated to be 91% by FDA.

#### 2.1.1. Effects of Enzymolysis Concentration and Duration on Protoplast Isolation

Enzymolysis liquid concentration and duration of enzymolysis are crucial factors affecting protoplast quantity and viability. To determine the optimal enzymolysis liquid concentration and duration of enzymolysis, protoplasts were isolated from the leaves by digestion using fifteen different enzymatic treatments for between 4 and 8 h. The prefabricated liquid in the enzyme solution of the fifteen different treatments was 0.7 M. The heatmap in Figure 2a provides a convenient visual overview of the results from the different treatments tested. The results indicate that with a combination of 1.25% Cellulase R-10 and 0.6% Macerozyme R-10, an optimal yield of 1.5 × 10^7^/g FW can be obtained after 5 h.

#### 2.1.2. Effects of Prefabricated Liquid Concentration on Protoplast Isolation

In maize, pretreatment of the samples with a balanced osmotic buffer prior to digestion significantly increases the efficiency of protoplasts generated [12]. Therefore, we applied a pretreatment buffer to the *U. rhynchophylla* samples for 20 min. The D-mannitol concentration in the prepared solution ranged from 0.5 to 1.0 M. All the D-mannitol treatments used 1.25% Cellulase R-10 and 0.6% Macerozyme R-10 for 5 h. When the D-mannitol concentration was increased to 0.7 M, a significant increase in yield and viability was observed. When the D-mannitol concentration was further increased to 0.8 M, the yield and viability were the highest and then sharply decreased at 1.0 M (Figure 2b). Figure 1b shows that the effect of preformed fluid concentration on protoplast production and viability is nonlinear, with an “inflection point”. These results show that 0.8 M D-mannitol produced the best results when combined with 1.25% Cellulase R-10 and 0.6% Macerozyme R-10 with a digestion time of 5 h.

#### 2.1.3. Effects of Temperature on Protoplast Isolation

As the enzymatic digestion temperature increased, protoplast production and viability increased and then decreased, with the highest protoplast production occurring at a temperature of 26 °C. Simultaneously, we found that the activity of protoplasts was significantly increased from 28 ± 2.2% to 86. ± 1.9%.

### 2.2. Protoplast Transformation

To optimize the protocol for the transformation of *U. rhynchophylla* protoplasts, several parameters were investigated based on previous reports, including the concentration of PEG, the amount of plasmid DNA, as well as the effect of incubation time and temperature [14]. Collected protoplasts were resuspended in an appropriate volume of MMG solution and transfection efficiency was evaluated using 2 × 10^5^ cells/mL protoplast.

It has been reported that the concentration of PEG used in protoplast transfection is a critical factor in determining transfection efficiency [28]. The optimal transformation efficiency was measured empirically in 20–60% PEG concentrations for each experimental condition. The 40% PEG concentration showed significantly higher transformation efficiency than other PEG concentrations (Figure 3a). We found that the amount of plasmid DNA used for protoplast transformation also influenced the transformation efficiency, as 40 μg of the 35S:GFP vector (Figure 3b) showed higher transformation efficiency reaching a maximum of 62.5% when added to the protoplasts. Accordingly, transforming 2 × 10^5^ cells/mL protoplasts in a 200 μL solution with 40 μg plasmid was found to be optimal. To further optimize transformation efficiency, transformation time was also examined (20, 30, 40, 50, and 60 min). The transformation temperature affected the efficiency of PEG-mediated plasmid transformation [29]. The transformation efficiency of protoplasts increases first and then decreased with the increase in temperature (Figure 3c). When the temperature was 4 °C, the activity of protoplasts remained stable, but the transformation efficiency was very low (Figure 4e–h). The optimal transformation temperature was found to be 24 °C, which is close to the optimal temperature identified for protoplast isolation. With 40 min of transformation time, the transformation efficiency was almost 71%, which was significantly higher than that of 30 min and 60 min, whereas the difference was not significant between 40 min and 50 min (Figure 3d). These results illustrate that under the conditions tested, the *U. rhynchophylla* protoplast transformation rate was highest (Figure 4a–d) when protoplasts were transformed overnight at 24 °C with 40 µg of plasmid DNA for 40 min in a solution containing 40% PEG.

The optimal conditions found for protoplast extraction, purification, and transformation of *U. rhynchophylla* tissue culture seedling leaves from the many varied conditions tested are summarized in Figure 5.

### 2.3. Subcellular Localization Analysis of UrWRKY37 Using U. rhynchophylla Protoplasts

In order to validate the feasibility of our protoplast system, we selected *U. rhynchophylla* TF, UrWRKY37, as a GFP fusion protein for the subcellular localization study. UrWRKY37 has the highest homology with CrWRKY1, and it has been reported that it can positively regulate the TIA biosynthesis pathway. According to the speculation, UrWRKY37 may have a similar function, so it became the object of our study. To determine the subcellular localization of UrWRKY37, we constructed a pBI121-GFP fusion protein vector pBI121-UrWRKY37-GFP. As mentioned above, we separated and transformed protoplasts using our optimal experimental conditions (Figure 5). The UrWRKY37 expression vector was successfully expressed in *U. rhynchophylla* protoplasts (Figure 6). As shown in Figure 6, the protoplast transformed with pBI121-GFP display fluorescent signals in the nucleus, cytosol, and cell membrane (Figure 6a), whereas the protoplast cell transfected with pBI121-UrWRKY1-GFP produced a green fluorescent signal only in the cell nucleus (Figure 6b), consistent with the predicted subcellular localization result and provide evidence for the practicability of the protoplast isolation and transient expression system we have developed.

### 2.4. Dual-Luciferase (Dual-LUC) Assays

Transient expression of luciferase (LUC) reporters in protoplast tobacco cells is widely used for assessing the activities of *cis*-regulatory elements [30]. Meanwhile, to further verify whether UrWRKY37 can bind to the *UrTDC* promoters, we performed a transient expression assay in *U. rhynchophylla* protoplasts. The *UrTDC* promoter regions were used to drive the luciferase gene (LUC) and Renilla luciferase (REN) driven by the 35S promoter as a fusion reporter, with UrWRKY37 overexpressed under the control of the 35S promoter as an effector (Figure 7a). UrWRKY37 could activate *UrTDC* transcription in *U. rhynchophylla* protoplasts; luciferase activity of the *UrTDC* was increased by *UrWRKY37* compared with the pHB vector significantly (Figure 7b). These results indicated that UrWRKY37 could bind to the promoters of the *UrTDC* and can activate LUC expression.

## 3. Discussion

### 3.1. Factors Influencing the Isolation of Protoplasts in U. rhynchophylla

In comparison to agricultural crops, there is a limited number of reports on protoplast cultures applied to medicinal plants. In this study, protoplasts were isolated from the leaves of *U. rhynchophylla* tissue culture seedlings with a growth period of 8 weeks (Figure 1a), and the separation conditions were optimized from the aspects of enzyme combination, digestion time, D-mannitol concentration, and temperature. The optimal protoplasts isolation protocol was obtained from results obtained from a wide variety of conditions tested. Overall, the optimal combination of enzymatic hydrolyzates was found to be 1.25% Cellulase R-10 and 0.6% Macerozyme R-10, 0.8 M prefabricated with enzymatic hydrolysis at 26 °C for 5 h (Figure 1d,e). Remarkably, the optimal enzyme concentration for protoplast isolation was found to be very close to that identified for *Cymbidium orchids* [2].

When exploring the effect of the combined concentration of enzymatic hydrolysate and the time of enzymatic hydrolysis on the protoplasts of *U. rhynchophylla*, the yield of protoplasts produced by 1.25% Cellulase R-10 and 0.6% Macerozyme R-10 was the highest, followed by 1.5% Cellulase R-10 and 0.6% Macerozyme R-10 (Figure 2a). With the increase in the concentration of the diastase enzyme, the number of protoplasts in the visual field increased but the number of intact protoplasts decreased. High enzyme concentration may adversely affect the integrity of the plasma membrane and the physiological and metabolic activities of protoplasts [31].

Plasmolysis during protoplast isolation is affected by the type and concentration of the osmoticum which helps to maintain the turgor pressure in the resulting protoplasts [32]. Therefore, the leaves of *U. rhynchophylla* were soaked in 0.8 M D-mannitol prefabrication solution for 20 min, and the results showed that the stability and activity of protoplasts after pretreatment were significantly increased from 51 ± 3% to 79 ± 1.4% (Figure 2b). When D-mannitol concentration increased from 0.5 M to 0.8 M, the yield and activity of protoplasts increased significantly, but further increases reduced the yield and activity of protoplasts. The decrease in the production of protoplasts at high concentrations may in part be due to the destruction of the integrity of the cell membrane by high concentrations of enzymes and the adverse effects on the physiological activities of protoplasts [33]. Osmotic conditions were found to significantly influence the yield of viable protoplasts, which supports previous reports [34].

In the process of protoplast separation, with increases in temperature, the yield of protoplast gradually increased and the digestion time corresponding to the maximum yield was shortened accordingly. However, high temperatures lead to a slow decrease of enzyme activity [35], and the activity of protoplast was found to also decrease. The yield and activity were found to be optimal at 26 °C for 5 h. Therefore, 26 °C is considered to be the most suitable temperature for enzymatic hydrolysis of *U. rhynchophylla* leaves (Figure 2c) and is similar to the optimal temperatures found in eggplants [14].

### 3.2. Factors Influencing the Transformation Efficiency of Protoplasts in U. rhynchophylla

The success of the transient expression protoplast system requires a suitable transformation method. PEG-mediated transfection has been widely used for the transient expression of many plant species, including ornamental plants [1,2,8], medicinal plants [36], edible fungi [37], and algae [38], and is also considered an effective protoplast-transforming agent that is widely used in many plant species [39]. Although the transfection efficiency increased with an increase in PEG4000 concentration and incubation period, the number of broken protoplasts increased after reaching a threshold concentration. High PEG4000 concentration and long incubation time can improve conversion efficiency. The results showed that the optimal protocol was incubation for 40 min with 40% of PEG4000 with transformation efficiency reaching 71% (Figure 3a,d). Therefore, proper PEG concentration and incubation time should be fully considered when establishing an effective protoplast transfection scheme.

Plasmid concentration is an important factor for the functional analysis of transfected protoplasts. For *U. rhynchophylla* protoplast transfection, the transformation rate was the highest when 200 μL of protoplast suspension was mixed gently with 40 μg of plasmid DNA. This concentration is consistent with the plasmid concentration used for the transfection of Chinese cabbage identified by Sivanandhan et al. [40]. During the transformation, we found that the efficiency of transformation increased with plasmid concentration. This result suggests that larger amounts of plasmid DNA could help increase the transformation efficiency [41]. Using plasmid quantities than larger 40 μg is practically possible, but it is important to vary the relative ratio between the quantity and quality of protoplast to achieve better transformation efficiency [42].

It has been shown that high-temperature heat shock can improve the transformation efficiency of potato protoplasts [43]. Our results on the effect of temperature on protoplast transformation showed that the lower the temperature, the better the survival of protoplasts, but with reduced plasmid transformation efficiency. When the transformation temperature was 24 °C, the transformation efficiency of the protoplasts reached 68%. When the temperature was 28 °C, more than 80% of the protoplasts died (Figure 3c).

### 3.3. Utilization and Validation of the Viability of the Protoplast System of U. rhynchophylla

The *U. rhynchophylla* protoplast isolation, purification, and transfection method outlined in Figure 5 provides a rapid and transient transformation method for analyzing the gene function of *U. rhynchophylla*. Using this system, we successfully studied the subcellular localization of UrWRKY37 (Figure 6). The results showed that UrWRKY37 was distributed in the nucleus of the protoplast of *U. rhynchophylla*, which is consistent with predicted subcellular results, indicating that the isolated protoplast is suitable for subcellular localization studies. In addition, the results of double luciferase detection showed that the expression of *UrTDC* (Figure 7b), a key gene related to alkaloid synthesis in *U. rhynchophylla*, was activated by the transcription factor UrWRKY37.

## 4. Materials and Methods

### 4.1. Plant Materials and Growth Conditions

The plant tissue derived from *U. rhynchophylla* was collected from the College of Horticulture, Hunan Agricultural University, and identified by Prof. Shugen Wei. Plant seedlings were grown for 60 days in tissue culture before being used. The media composition for plant tissue culture was 1/2 MS medium, with 3-indolebutyric acid: 0.2 mg/L; 1-naphthlcetic acid: 0.2 mg/L; sucrose: 25 g/L; agar: 4.5 g/L; and activated carbon: 0.5 g/L. Seedlings were cultivated in the greenhouse at 25 °C under a 12/12 h (light/dark) condition with a light intensity of 120–150 μmol/m^2^/s and 60–70% relative humidity for 2 months.

### 4.2. Plasmid Construction

Total RNA was extracted from leaves of *U. rhynchophylla* using the SteadyPure Plant RNA Extraction Kit (Accurate Biology, Hunan, China). The purity and concentration of RNA were measured by Micro Drop (BIO-DL, Shanghai, China), and the RNA integrity was checked on 1% agarose gels. Total RNA (500 ng) was used for reverse transcription with Evo *M-MLV* RT Mix Kit with gDNA Clean (Accurate Biology, Hunan, China) to remove genomic DNA contamination for qPCR in a 20 μL volume according to the manufacturer’s instructions.

The pBI121-GFP vector, carrying a CaMV 35S promoter, was selected for characterizing the transformation efficiency of *U. rhynchophylla* protoplasts. To verify the feasibility of detecting the subcellular location of a *U. rhynchophylla* protein, the complete coding sequences of *UrWRKY37* with no stop codons including the restriction enzyme sites *Nde*l and *Sma*I were amplified using the primers 5′-GACAGTGACATATCCAAGAGACAGA-3′ (forward primer) and 5′-GCCAGCGATAGCCATCATTTAC-3′ (reverse primer) and cloned into the pBI121-GFP vector, establishing the expression plasmid pBI121-*UrWRKY37-GFP.* Plasmid DNA was extracted using the Plasmid MaxiPrep Kit (endotoxin-free) (Coolaber, Beijing, China). The DNA concentration was at least 500 ng/μL. All plasmid DNA were stored at −20 °C after confirmation.

### 4.3. Preparation of Enzyme Solution and Prefabricated Solution

To investigate the optimum enzymatic digestion solution concentration and duration of enzymatic digestion of *U. rhynchophylla* protoplast extraction, 15 concentrations of enzymatic solution were used to digest leaf squares in 4–8 h. The effects of temperature (20, 22, 24, 26, 28, and 30 °C) and prefabricated liquid (0.3, 0.4, 0.5, 0.6, 0.7, and 0.8 M) on the yield and viability of *U. rhynchophylla* protoplasts was also measured. The enzymolysis solution was prepared by adding different concentrations of Cellulase R-10 (Yakult, Tokyo, Japan), Macerozyme R-10 (Yakult, Tokyo, Japan), and 1.0 g M-mannitol (final 0.6 M); adding 500 μL of 2 × CPW solution (Table 1) and 1 mL 5 mmol/L MES (final 5 mM); dissolving by adding 5 mL DEPC treated water; and then vortexing for 2 min. The solution was heated to 55 °C for 10 min and then cooled to room temperature before adding 500 μL 5 M CaCl_2_ (final 2.5 mM) and 1 mL of 1% BSA (final 1%). DEPC-treated water was added to bring the volume to 10 mL (Table 2).

Different amounts of D-mannitol were weighed and dissolved in DEPC-treated water to configure prefabricated liquids at concentrations of 0.5, 0.6, 0.7, 0.8, 0.9, and 1.0 M, respectively. Both the enzyme solutions and the prefabricated liquid concentration were adjusted to pH 5.8, filtered and sterilized by 0.22 μm syringe filter, and then stored at 4 °C.

### 4.4. Protoplast Isolation

Expanded and healthy leaves (0.2 g) were collected from 8-week-old plants grown on ½ MS solid media plates and cut into small pieces of 0.5–1.0 mm with scissors. The leaf squares were quickly and gently dipped into the D-mannitol buffer concentration, making sure both sides of the squares were well emerged with the solution for 20 min, after which the prefabricated liquid was discarded. Then 10 mL of enzyme digestion solution was added into centrifuge tubes wrapped in tin foil that were constantly shaken at 25 °C. The enzyme solution turned green, indicating the release of protoplasts into the solution. An equal volume of ice-cold W5 solution (Table 3) was added to the protoplast suspension in the ultra-net workbench, mixing well through gentle shaking. A 75 μm nylon mesh was then used to remove undigested leaf tissues and the filtrate was dispensed into 50 mL sterile centrifuge tubes. As much supernatant was removed as possible, and the protoplast pellet was resuspended with an equal volume of ice-cold W5 solution by gentle swirling. The tube with resuspended W5 and the protoplast suspension was then put on ice in the dark for 30 min. Healthy protoplasts settled down at the bottom of the tube by gravity after 30 min incubation on ice.

### 4.5. Protoplast Transformation

PEG-mediated *U. rhynchophylla* protoplast transformation was performed according to the protoplast transformation of the eggplant lab protocol with some minor modifications [14]. Before the transformation, isolated protoplasts were kept on ice containing the W5 washing solution for 30 min. Protoplasts were allowed to settle to the bottom of the tube, and then as much washing solution was removed as possible without touching the protoplast pellet. Protoplast pellets were then resuspended in MMG solution (Table 4) to obtain densities of 2.0–3.0 × 10^5^ cells/mL. The pH of the solution was then adjusted to 5.8. A volume of 20 μL pBI121-UrWRKY37-GFP plasmid at different DNA amounts (5, 10, 20, 30, 40, and 50 μg) was added to 200 μL of the prepared protoplasts and mixed gently. An equal volume (220 μL) of freshly prepared PEG4000 solution (Table 5) was then immediately added, and the suspension was carefully mixed by gently inverting the tube. To optimize PEG concentration, different concentrations of PEG4000 (20, 30, 40, 50, and 60%, *w*/*v*) were tested. To optimize the transformation duration, the mixture was incubated for 20, 30, 40, 50, and 60 min in the dark at room temperature. To optimize the transformation temperature, different temperatures (4, 20, 22, 24, 26, and 28 °C) were also tested.

The transformed protoplast mixture was washed with 1 mL of chilled W5 solution, followed by centrifugation at 50 g for 2 min to stop the reaction. As much of the supernatant was removed as possible without touching the protoplast pellet. All the samples were then incubated in a dark environment at 24 °C for 10–14 h in illuminated incubators before microscopic examination.

### 4.6. Protoplast Yield and Viability

The protoplast yield was determined under an optical microscope (Olympus, Tokyo, Japan). It was expressed as the number of protoplasts/g FW as follows: Y = n × 10^5^ × p/g FW, where Y is the protoplast yield (protoplasts/g FW), n is the number of protoplasts/mL, p is the protoplast suspension volume (mL), and g FW is the leaf one-gram fresh weight.

Protoplast viability was determined by staining with 0.01% (w/v) fluorescein diacetate (FDA). The protoplast suspension (100 μL) was stained with FDA (1 μL) and incubated in the dark for 5 min. Observations were made under a fluorescence microscope. Protoplast viability (%) = (number of the fluorescent protoplast in view/number of the total protoplasts in view) × 100%. During transformation, the number of total cells and fluorescent cells was measured. All fluorescence experiments were independently repeated at least three times with similar results. Transfection efficiency (%) = (fluorescent protoplasts in view/total protoplasts in view) × 100%. All data are shown as the mean of three independent experiments with a significance level, and figures were created using TBtools v.v 106 and GraphPad Prism 9. 0. 0.

### 4.7. Subcellular Localization

The subcellular localization of the transcription factor UrWRKY37 in the protoplasts of *U. rhynchophylla* was carried out according to the best method determined from the results of our optimization experiments. The subcellular localization was predicted by Utilize WoLF PSORT (http://www.genscript.com/wolf-psort.html, accessed on 1 November 2022).

### 4.8. Dual-Luciferase Assays

Tryptophan decarboxylase (TDC) can convert tryptophan into tryptamine, which is a key enzyme involved in the rhynchophylla biosynthesis pathway. In order to discover whether UrWRKY37 can activate *UrTDC* promoters, the coding sequences of *UrWRKY37* were cloned and constructed into a pHB vector (Gene create, Wuhan, China) as an effector. The promoter sequences of *UrTDC* were cloned into the pGreenII 0800 vector (Gene create, Wuhan, China) containing Renilla luciferase (REN) and firefly luciferase (LUC) genes as the reporter. The pHB empty vector was used as a negative control, and the REN gene driven by the 35S promoter was used as an internal reference. The 0.2 μg of the reporter and effector plasmids co-transferred into *U. rhynchophylla* protoplasts. Luciferase activity was measured using the Dual-Luciferase^®^ Reporter Assay System (Promega, Madison, America) 48 h post-transfection. The binding ability of UrWRKY37 to the *UrTDC* promoter is indicated by the LUC/REN activity ratio. All the experiments were repeated five times.

### 4.9. Microscopy and Bioimage

Fluorescence signals of protoplasts were observed using a fluorescent confocal microscope (Zeiss Apotome 2, Jena, Germany) with Bright, GFP (486 nm) mRF12 (449 nm), and mCherry (587 nm), respectively. FDA was stimulated at 486 nm wavelength. All fluorescence experiments were repeated independently at least three times.

## 5. Conclusions

In this study, we established an efficient and reliable protoplast isolation and transient expression system for *U. rhynchophylla* after testing many different concentrations that have been found to affect the efficiency of other plant species. An optimized protocol for *U. rhynchophylla* protoplast isolation, purification, and transient gene expression was developed. The protoplast yield was as high as 1.5 × 10^7^ protoplasts/g FW, and protoplast viability was 91%. The *U. rhynchophylla* protoplast transfection rate was highest (71%) when protoplasts were transfected overnight at 24 °C with the 40 µg of plasmid DNA for 40 min in a solution containing 40% PEG. The UrWRKY37 was localized in the nucleus and dual-luciferase (Dual-LUC) assays indicated UrWRKY37 could bind to the promoters of *UrTDC* and activate its expression. Here, the protoplast system may broadly provide insight into the studies of *U. rhynchophylla*, including scRNA-seq, genome editing, protein localization, protein–protein interactions, and gene function identification.

## Figures and Tables

**Figure 1 ijms-24-03633-f001:**
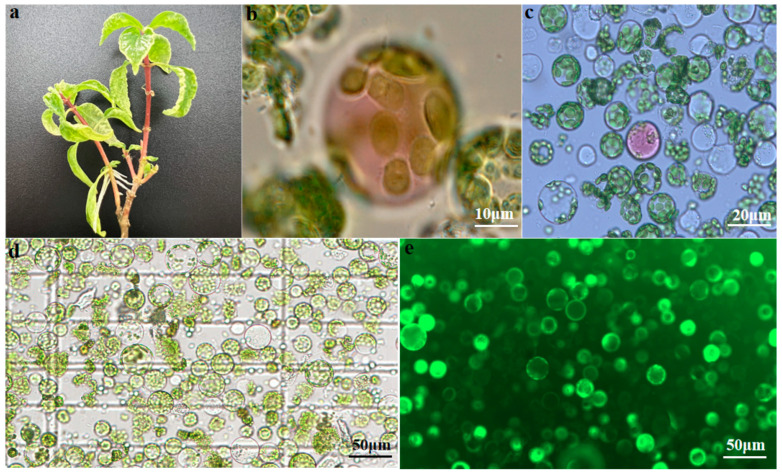
Isolation of protoplasts from leaves of *U. rhynchophylla* plantlets. (**a**) Initial material used for protoplast isolation. (**b**) Freshly isolated protoplasts, bar = 10 µm. (**c**) Freshly isolated protoplasts, bar = 20 µm. (**d**) The yield of *U. rhynchophylla* protoplasts was observed under a fluorescence microscope with ×10 objective, bar = 50 μm; bright field channel. (**e**) The viability of *U. rhynchophylla* protoplasts was observed under a fluorescence microscope with a × 10 objective. Protoplasts with green fluorescence were viable. Bar = 50 μm. GFP channel.

**Figure 2 ijms-24-03633-f002:**
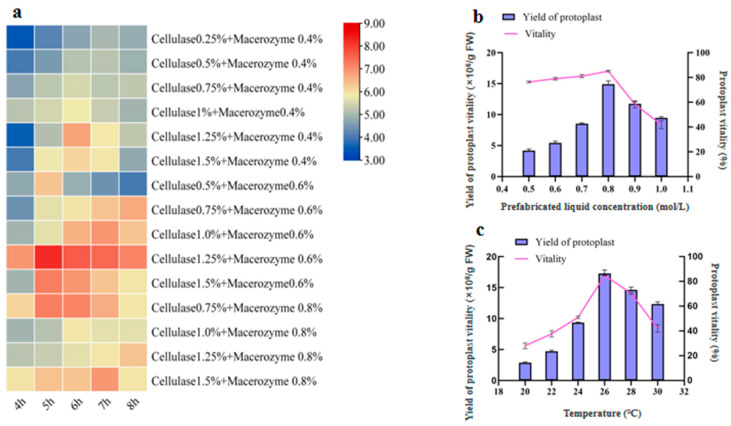
Factors influencing the isolation of *U. rhynchophylla* protoplasts. (**a**) The heatmap of *U. rhynchophylla* protoplast yield under different enzyme concentrations and digestion times, the darker the red color, the higher the yield of protoplasts. (**b**) Effect of different D-mannitol concentrations on the yield and viability of *U. rhynchophylla* protoplasts. (**c**) Effects of different temperatures on the yield and viability of *U. rhynchophylla* protoplasts.

**Figure 3 ijms-24-03633-f003:**
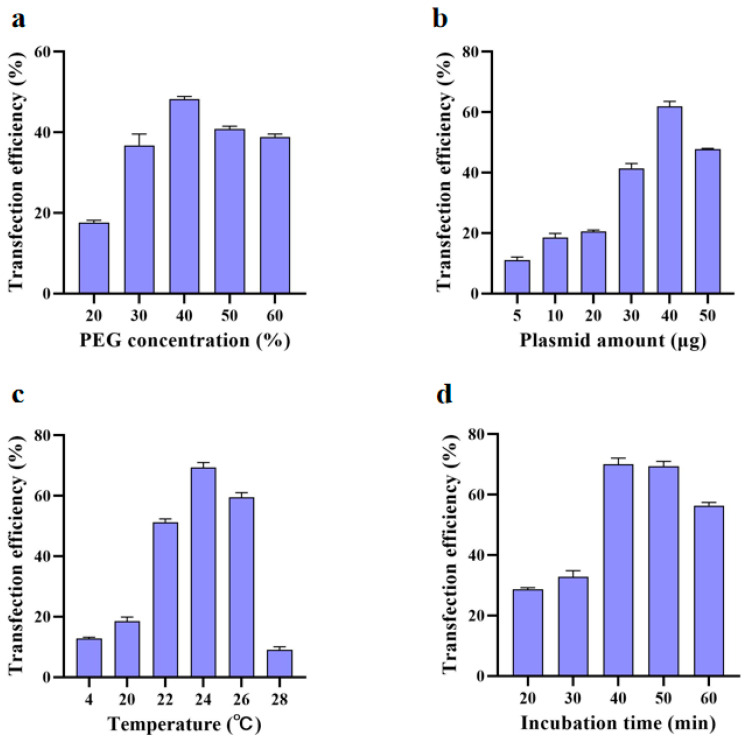
Transformation efficiency of *U. rhynchophylla* protoplasts. (**a**) Effects of PEG concentration on the efficiency of transient expression. (**b**) Effects of plasmid amount on transfection efficiency. (**c**) Effects of temperature on transformation efficiency. (**d**) Effects of incubation time on transformation efficiency.

**Figure 4 ijms-24-03633-f004:**
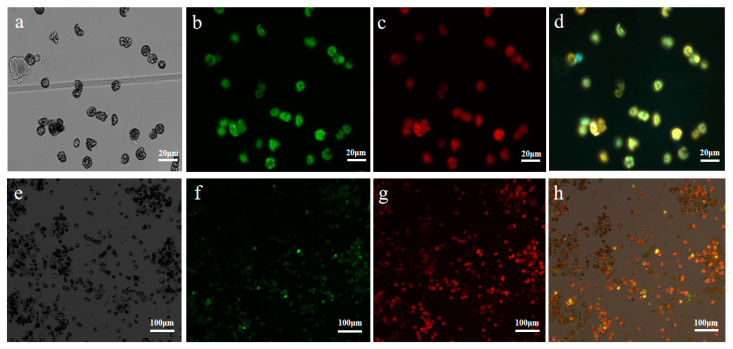
Transient *UrWRKY37* gene expression in *U. rhynchophylla* protoplasts observed under a fluorescence microscope. (**a**–**d**) The best transfection efficiency of the GFP construct was observed under a fluorescence microscope with a × 20 objective at 24 °C, bars = 20 μm. (**e**–**h**) The transfection efficiency of the GFP construct was observed under a fluorescence microscope with a × 5 objective at 4 °C, bars = 100 μm. (**a**,**e**) Protoplast with bright field (BF). (**b**,**f**) Green fluorescent protein (GFP). (**c**,**g**) Chlorophyll autofluorescence (Chl). (**d**,**h**) Merged images were simultaneously exhibited.

**Figure 5 ijms-24-03633-f005:**
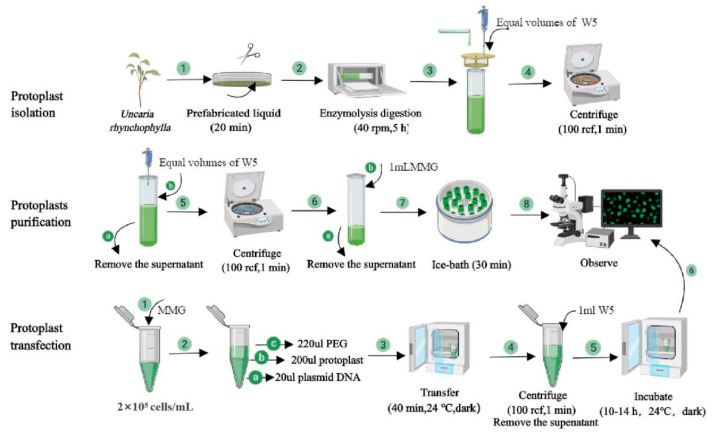
Schematic diagram of extraction, purification, and transformation process of *U. rhynchophylla* protoplasts.

**Figure 6 ijms-24-03633-f006:**
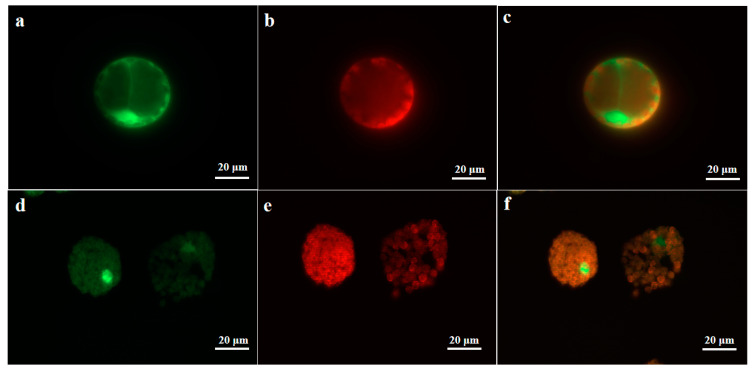
The subcellular localizations of UrWRKY37 using *U. rhynchophylla* protoplasts. Subcellular localization of pBI121-GFP (**a**–**c**) empty vectors and pBI121-UrWRKY37-GFP (**d**–**f**) in *U. rhynchophylla* protoplasts under a fluorescence microscope. (**a**,**d**) Protoplast with GFP. (**b**,**e**) Chlorophyll autofluorescence (Chl). (**c**,**f**) Merged images were simultaneously exhibited. Bars = 20 μm.

**Figure 7 ijms-24-03633-f007:**
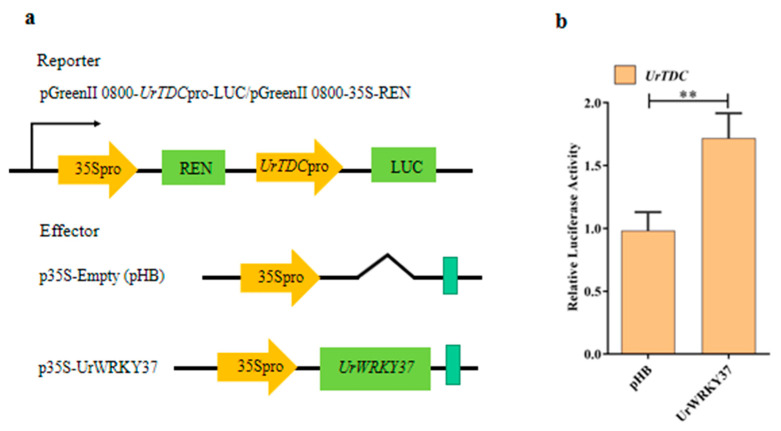
Activation of UrTDC promoter by UrWRKY37 in dual-luciferase (Dual-LUC) assay. (**a**) Schematic diagram of the reporter and effector plasmids used in the Dual-LUC assay. The UrTDC promoter sequence was inserted into pGreenII 0800 vector containing LUC and Renilla luciferase (REN) as the reported plasmid. The effector pHB plasmid containing UrWRKY37 coding sequences driven by the 35S promoter. The binding ability of UrWRKY37 to the UrTDC promoter is indicated by the LUC/REN ratio. (**b**) The promoter activity of the UrTDC promoter is expressed as the LUC/REN ratio. Error bars indicate the standard error of five biological replicates. ** denotes a statistically remarkable difference.

**Table 1 ijms-24-03633-t001:** Composition of the 2 × CPW solution (1 L).

Reagent	Amount
KH_2_PO_4_	54.4 mg
KNO_3_	202.0 mg
CaCl_2_2H_2_O	2960.0 mg
MgSO_4_	492 mg
KI	0.32 mg
CuSO_4_	0.05 mg
ddH_2_O	Up to 1 L

**Table 2 ijms-24-03633-t002:** Composition of enzymatic hydrolysate (10 mL).

Reagent	Final Concentration	Amount
Cellulase R-10	1.25%	0.125 g
Macerozyme R-10	0.60%	0.6 g
D-mannitol	0.6 M	1.092 g
2-CPW	-	500 μL
5 mM MES	5 mM	1 mL
DEPC-treated water	5 mL	
55 °C water bath pot	Heating the solution at 55 °C for 10 min and then cool it to room temperature	
5 M CaCl_2_	2.5 M	500 μL
1% BSA	0.10%	1 mL
DEPC treated water	Up to 10 mL and the PH of the enzyme solution was 5.8	

**Table 3 ijms-24-03633-t003:** Composition of W5 buffer (50 mL).

Reagent	Final Concentration	Amount
2 M KCl	5 mM	0.125 mL
1 M CaCl_2_	125 mM	6.25 mL
0.5 M NaCl	154 mM	15.4 mL
0.2 M MES	2 mM	0.5 mL
ddH_2_O		up to 50 mL

**Table 4 ijms-24-03633-t004:** Composition of MMG buffer (5 mL).

Reagent	Final Concentration	Amount
0.8 M D-mannitol	0.6 M	3.75 mL
2 M MgCl_2_	15 mM	37.5 mL
0.2 M MES	4 mM	0.1 mL
ddH_2_O	-	up to 5 mL

**Table 5 ijms-24-03633-t005:** Composition of 40% PEG solution (1 mL).

Reagent	Final Concentration	Amount
PEG4000	40% (*m*/*v*)	0.4 g
0.8 M D-mannitol	0.2 M	250 μL
1 M CaCl_2_	0.2 M	200 μL
ddH_2_O	-	up to 1 mL

## Data Availability

Not applicable.

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
