# Peer review of "Optimization of Isolation and Transformation of Protoplasts from Uncaria rhynchophylla and Its Application to Transient Gene Expression Analysis"

_ijms, 2023, doi:10.3390/ijms24043633_

Round 1

Reviewer 1 Report

In this manuscript, the authors present a protocol for protoplast isolation and transformation in Uncaria rhynchophylla. Uncaria rhynchophylla is known as a medicinal plant owing to its production of alkaloids, some of which have potential use in the treatment of neurological diseases.

The protocol developed in this manuscript is specific to this plant. Also, some experiments conducted in this study were not well-presented or designed. Therefore, clear explanations and support from new experiment are needed before re-consideration for publication.

1. It is better to provide the bright field image of protoplasts of Figure 1E, as it is difficult to assess the viability of protoplasts without displaying the total number of protoplasts.

2. In Line 162-164, the authors stated that "Increasing the temperature above ambient temperature, increased the number of transformed cells, but the overall viability of protoplasts was found to decrease dramatically (Figure 4b)." However, the content in Figure 4b is irrelevant. The figure legend of Figure 4 should also be revised to clearly state the difference between a-d and e-h.

3. Please provide the bright field image in Figure 6.

4. The authors should explain the reason behind choosing UrWRKY37 in this study.

5. The authors cloned UrWRKY37 into pcDNA3.1 for the dual luciferase assay. pcDNA3.1 is widely used as a mammalian expression vector but not a plant expression vector. To my knowledge, it does not contain any plant promoter. Although the increase in relative luciferase activity recorded in Figure 7b is less than 50%, which is much less than that usually reported in a typical system (e.g. driven by 35S promoter), the author still claimed that transactivation activity was observed using this system. The authors should design a valid system and experiment again to demonstrate the reliability of their protocol.

6. Endotoxins remained from plasmid preparation may also affect protoplast yield and viability. The authors should consider optimizing the protocol with the endotoxin removal step.

Reviewer 2 Report

Manuscript entitle: Optimization of Isolation and Transformation of Protoplasts from Uncaria rhynchophylla and Its Application to Transient Gene Expression Analysis by

 Yingying Shao et al.

The overall manuscript is very interesting and clearly shows the relevance of obtaining the protoplasts from Uncaria rhynchophylla, a plant highly demanded worldwide by the pharmaceutical industry for the indole alkaloids that it produces.

To increase the alkaloids production eventually, authors generated U. rhynchophylla protoplast isolation, purification, and transient gene expression from this plant.

This paper also mentions the relevance and the medical application of the chemical compounds of Uncaria rhynchophylla. The method used to generate protoplasts will be useful for future research to elucidate gene function in U. rhynchophylla. The manuscript is original and is the first time all the information about protoplast and transient transformation is carried out in this plant.

The paper is well-documented and well-written, and the figures and tables show the information clearly and are interesting.

Thus, I have a few minor general comments to improve the structure of the review.

I suggest introducing subtitles in the discussion and going more deeply into each section to cover all aspects of the results obtained.

Author Response

The overall manuscript is very interesting and clearly shows the relevance of obtaining the protoplasts from Uncaria rhynchophylla, a plant highly demanded worldwide by the pharmaceutical industry for the indole alkaloids that it produces. To increase the alkaloids production eventually, authors generated U. rhynchophylla protoplast isolation, purification, and transient gene expression from this plant. This paper also mentions the relevance and the medical application of the chemical compounds of U. rhynchophylla. The method used to generate protoplasts will be useful for future research to elucidate gene function in U. rhynchophylla. The manuscript is original and is the first time all the information about protoplast and transient transformation is carried out in this plant. The paper is well-documented and well-written, and the figures and tables show the information clearly and are interesting. Thus, I have a few minor general comments to improve the structure of the review.

  1. I suggest introducing subtitles in the discussion and going more deeply into each section to cover all aspects of the results obtained.

Response: Thanks for your kind comments. The authors have introduced subtitles in the discussion.

3.1. Factors influencing the isolation of protoplasts in U. rhynchophylla

3.2. Factors influencing the transformation efficiency of protoplasts in U. rhynchophylla

3.3. Utilization and validation of the viability of the protoplast system of U. rhynchophylla

Round 2

Reviewer 1 Report

The authors have responded to the suggestions and conducted new experiments. Before publicaiton, the following comments should be considered.

1. The language in Line 86-Line 90 needs to be polished.

2. Although it was stated in the main text, please also indicate the optimal temperature in the notes of Figure 4(a-d).

3. Please clarify the information of the PHB vector and provide reference. The meaning of "35S-PHB-UrWRKY37" is confusing. Was "PHB" placed between 35Spro and UrWRKY37?
